# Cu-Catalyzed Oxidative 3-Amination of Indoles via Formation of Indolyl(aryl)iodonium Imides Using *o*-Substituted (Diacetoxyiodo)arene as a High-Performance Hypervalent Iodine Compound

**DOI:** 10.3390/molecules24061147

**Published:** 2019-03-22

**Authors:** Kazuhiro Watanabe, Katsuhiko Moriyama

**Affiliations:** Department of Chemistry, Graduate School of Science, and Molecular Chirality Research Center, Chiba University, 1-33 Yayoi-cho, Inage-ku, Chiba 263-8522, Japan; cars9s66@chiba-u.jp

**Keywords:** *o*-substituted iodoarene, indole, Cu catalyst, C–N coupling reaction, hypervalent iodine, one-pot reaction, regioselectivity

## Abstract

An oxidative 3-amination of indole derivatives using hypervalent iodine(III) and bissulfonimides, which proceeds via the formation of indolyl(aryl)iodonium imides, was developed. This reaction was followed by an indole-selective copper-catalyzed oxidative C–N coupling reaction to obtain 3-amino indole derivatives as single regioisomers. *o*-Alkoxy(diacetoxyiodo)arenes showed higher reactivity in the reaction than *o*-alkyl(diacetoxyiodo)arenes, efficiently promoting the formation of indolyl(aryl)iodonium imides in the first step.

## 1. Introduction

The direct installation of an amino group into an indole skeleton is a very important strategy to generate amino indole derivatives, which are key building blocks for indole-based natural products and biologically active compounds [1,2,3,4,5,6]. The novel 3-selective amination of indole derivatives has attracted considerable attention from synthetic organic chemists. Regarding previous work on the 3-selective amination of indole derivatives, the nitration of protecting-group-free and *N*-protected indoles with metal nitrate or nitric acid is well known (Scheme 1A) [7,8,9,10,11,12]. Sudalai and coworkers reported an azidation of indole derivatives with a stoichiometric amount of I_2_ and NaN_3_ to give 3-azide indole derivatives (Scheme 1B) [13]. Wang and coworkers developed an electrophilic amidation of indoles with *N*-[(benzenesulfonyl)oxy]amide as the electrophilic nitrogen agent in the presence of ZnCl_2_ (Scheme 1C) [14]. As examples of the C–N coupling reaction of indoles with a nucleophilic nitrogen source, the azidation with NaN_3_ via the formation of indolyl(phenyl)iodonium azide (Scheme 1D) [15], and the amination of alkylamines via the formation of indolyl(Mes)iodonium tosylate (Scheme 1E) [16] from electron-donating group protected 2-substituted indoles using hypervalent iodine compounds were developed by Suna and coworkers. Regarding the 3-amination of electron-withdrawing group protected indoles, the 3-imidation of *N*-acetyl indoles with (PhSO_2_)_2_NF under transition-metal-free conditions was developed by Yang and coworkers (Scheme 1F) [17].

On the other hand, we were able to prepare indolyl(aryl)iodonium imides, which are related to I–N bonding hypervalent iodine compounds, from *N*-electron-withdrawing group protected indoles, bissulfonimides, and (diacetoxyiodo)arenes [18,19], and to achieve the dual functionalization of indoles via halo-amination with halogen reagents using the indolyl(aryl)iodonium imides [18,20].

We report herein a 3-selective C–N coupling reaction of *N*-electron-withdrawing group protected indoles with various bissulfonimides using hypervalent iodine compounds (Scheme 1G). In our previous work, the indole-selective C–N coupling reaction of indolyl(aryl)iodonium imides promoted by a copper catalyst gave 3-amino indole derivatives [21]. Furthermore, in the copper-catalyzed C–N coupling reaction, indolyl(2-butoxyphenyl)iodonium bistosylimide showed high indole selectivity whereas indolyl(phenyl)iodonium bistosylimide showed aryl selectivity (Scheme 2). From the results, we envisioned that (diacetoxy)iodoarene bearing an *o*-substituted phenyl group would promote the copper-catalyzed C–N coupling reaction of indoles with bissulfonimides via the formation of indolyl(aryl)iodonium imides. To obtain the corresponding 3-amino indole derivatives using this method, the formation of indolyl(aryl)iodonium imides should proceed effectively other than indole selectivity in the C–N coupling reaction of indolyl(aryl)iodonium imides. However, it is possible that steric hindrance by *o*-substituted bissulfonimidoiodoarene (ArI(OAc)(N(SO_2_R^2^)_2_) or ArI(N(SO_2_R^2^)_2_)_2_), which is generated in situ from the reaction of (diacetoxyiodo)arene with bissulfonimides, would suppress the formation of indolyl(aryl)iodonium imides. The necessity of appropriate steric hindrance and the electronic effect of *o*-alkoxy group substituted iodoarene moiety on hypervalent iodine(III) compounds in the 3-amination of *N*-electron-withdrawing group protected indoles were evaluated on the basis of comparisons with other *o*-substituted (diacetoxyiodo)arenes.

## 2. Results and Discussion

First, we conducted screening for (acetoxy)iodoarene, copper catalyst, and solvent for the 3-amination of indoles on the basis of previously reported reaction conditions (Table 1). It is important to use MeCN as a solvent in the formation of indolyl(aryl)iodonium imides (in the first step) and toluene-related solvent (i.e., toluene and xylene) in the copper-catalyzed C–N coupling reaction (in the second step) to progress each reaction efficiently [18,20,21]. Treatment of *N*-pivaloyl indole (**1a**) with Ts_2_NH (1.6 equiv.) and PhI(OAc)_2_ (1.2 equiv.) in MeCN at 40 °C for 7 h, followed by the addition of Cu(MeCN)_4_BF_4_ (5 mol%) in toluene under reflux conditions for 1 h provided *N*-pivaloyl-3-bistosylimido indole (**2a**) in 26% yield together with *N*,*N*-bistosylaniline in 58% yield as a byproduct (Entry 1). When (diacetoxyiodo)2-butoxybenzene (**A**), (diacetoxyiodo)2,4,6-trimethylbenzene (**B**), and (diacetoxyiodo)2,6-dimethoxybenzene (**D**) were used instead of PhI(OAc)_2_, the C–N coupling reaction showed high indole selectivity to obtain the desired product (**2a**) in high yields (76–82% yield) (Entries 2, 3, and 5). The use of (diacetoxyiodo)2,4,6-triisopropylbenzene (**C**) and 1-acetoxy-1,2-benziodoxol-3-(1H)-one (**E**) gave **2a** in moderate to low yields (Entries 4 and 6). (Diacetoxyiodo)arenes (**A**, **B**, and **D**) as *o*-substituted hypervalent iodine compounds were adopted for use in xylene at 130 °C for the 3-amination of indoles, and similar reactivity to that in toluene under reflux conditions was realized (Entries 7–9). By contrast, the reaction with (diacetoxyiodo)arenes **A**, **B**, and **D** in the absence of copper catalyst gave **2a** in low yields (16–42% yield) (Entries 10–12).

Then, (diacetoxyiodo)arenes **A**, **B**, and **D** were utilized in the reaction with *N*-pivaloyl 5-chloroindole (**1b**) under two conditions (in toluene under reflux conditions and in xylene at 130 °C) to clarify substrate generality for the 3-amination of indoles (Scheme 3). Treatment of **1b** with (diacetoxyiodo)2-butoxybenzene (**A**) in xylene at 130 °C in the second step provided the desired product (**2b**) in 75% yield. However, when (diacetoxyiodo)2,4,6-trimethylbenzene (**B**) was used under similar conditions to those in the reaction with **A**, the reactivity of **B** with **1b** declined to give **2b** in 48% yield. (Diacetoxyiodo)2,6-dimethoxybenzene (**D**) showed similar reactivity to **A** (68% yield). It is noteworthy that the reaction of **1b** in toluene under reflux conditions in the second step using the three (diacetoxyiodo)arenes (**A**, **B**, and **D**) generated small quantities of 2-aminated indole derivative as byproduct (5–9% yield). Therefore, we determined that the reaction in xylene at 130 °C in the presence of Cu(MeCN)_4_BF_4_ catalyst was the best option for the second step in the 3-amination of indoles using (diacetoxyiodo)arenes.

To compare the reactivities of (diacetoxyiodo)arenes (**A**, **B**, and **D**) in the copper-catalyzed 3-amination of indoles, *N*-protected indoles (**1**) were examined under optimum conditions (Scheme 4). Treatment of *N*-pivaloyl indole (**1a**) with various bissulfonimides, containing an aromatic group, an aliphatic group, or a dissymmetric functional group provided corresponding 3-amino indole derivatives (**2c**–**f**) in 51–81% yields, respectively. Unfortunately, the reaction with Ms_2_NH gave *N*-pivaloyl-2-dimesylimido-indole derivative (**3f**) in 10–14% yields as the by-product. The reaction of 5- and 6-substituted indoles bearing an alkyl group (**1g** and **1o**), a halogen (**1h**, **1i**, and **1p**–**r**), an ether (**1j** and **1s**), an ester (**1k** and **1l**), and a phthalimide (**1n**) also produced corresponding products (**2g**–**l**, **2n**–**s**) in 39–84% yields, respectively. Use of 7-substituted indole (**1u**) and 2-substituted indole (**1w**) derivatives in the present reaction also furnished the desired products (**2u** and **2w**) in good yields (60–76%). Other indole protecting groups, such as benzoyl (**1x**) and tosyl (**1y**) groups, instead of a pivaloyl group also enhanced the reactivities of (diacetoxyiodo)arenes in the present reaction to give 3-amino indole derivatives (**2x** and **2y**) in high yields (72–91%). In the reaction of 5-cyano (**1m**), 4-fluoro (**1t**), and 5,6-dichloro indole (**1v**) derivatives, (diacetoxyiodo)2-butoxybenzene (**A**) or (diacetoxyiodo)2,6-dimethoxybenzene (**D**) improved the yields of 3-amino indole derivatives (**2m**, **2t**, and **2v**) to 50–59% yield, whereas other (diacetoxyiodo)arenes showed low reactivity (20–36% yield). It is noteworthy that (diacetoxyiodo)arenes **A** and **D** are superior to (diacetoxyiodo)2,4,6-trimethylbenzene (**B**) in the one-pot 3-amination of indoles.

Furthermore, we examined the reaction of *N*-pivaloyl-5-chloroindole (**1b**) with Ts_2_NH and (diacetoxyiodo)arene (**A**, **B**, and **D**) in MeCN at 40 °C, which forms indolyl(aryl)iodonium imide (**4b**) (in the first step), to elucidate the reason why the reactivities of (diacetoxyiodo)arenes differ in the 3-amination of indole derivatives (Scheme 5). When (diacetoxyiodo)2-butoxybenzene (**A**) and (diacetoxyiodo)2,6-dimethoxybenzene (**D**) were used in this reaction, indolyl(aryl)iodonium imide (**4b**) was obtained in good yields (54% and 65%) together with 3-amino indole derivative (**2b**) (14% and 19%) and 2-amino indole derivative (**3b**) (12% and 13%). In contrast, (diacetoxyiodo)2,4,6-trimethylbenzene (**B**) showed lower reactivity than ArI(OAc)_2_
**A** and **D** (25% yield of **4b** and 15% recovery of **1b**). From the results, we suggest that the difference in reactivity among the (diacetoxyiodo)arenes in the 3-amination of indoles is strongly dependent on the reactivity in the formation of indolyl(aryl)iodonium imide in the first step.

In addition, it is possible that **3b** is converted into **2b** via rearrangement by a copper catalyst, based on the results that **3b** was not generated in xylene at 130 °C, but was obtained in toluene under reflux conditions (Scheme 3). The reaction of *N*-pivaloyl 2-bistosylimido-5-chloroindole (**3b**) in the presence of Cu(MeCN)_4_BF_4_ catalyst in xylene at 130 °C provided a complex mixture without the formation of **2b** (Scheme 6).

Based on the results of mechanistic studies, we propose a reaction mechanism for the 3-amination of indole derivatives, as shown in Scheme 7. The reaction of indole derivatives (**1**) with (diacetoxyiodo)arenes (ArI(OAc)_2_) and bissulfonimides ((R^2^SO_2_)_2_NH) in MeCN, which generates (sulfonimidoiodo) arene intermediates (ArI(OAc)(N(SO_2_R^2^)_2_) or ArI (N(SO_2_R^2^)_2_)_2_) in situ, provides indolyl(aryl)iodonium imides (**4**) together with 3-amino indole derivatives (**2**) and 2-amino indole derivatives (**3**) in the first step. The alkoxy groups at *o*-positions of iodoarenes result in higher reactivity than the alkyl groups to induce the formation of amino group-containing products. Subsequently, indolyl(aryl)iodonium imides (**4**) are converted into 3-amino indole derivatives (**2**) through an oxidative C–N coupling reaction using Cu(MeCN)_4_BF_4_ catalyst [21]. It is noteworthy that the *o*-substituent on iodoarene is essential to prompt the indole selectivity for the C–N coupling reaction of **4**. In contrast, 2-amino indole derivatives (**3**) are decomposed in the presence of a copper catalyst under heating conditions.

## 3. Materials and Methods

### 3.1. General Procedure

^1^H NMR spectra were measured on a JEOL ECS-500 (500 MHz) spectrometer at ambient temperature. Data were recorded as follows: chemical shift in ppm from internal tetramethylsilane on the δ scale, multiplicity (s = singlet; d = doublet; t = triplet; q = quartet; sep = septet; m = multiplet; br = broad), coupling constant (Hz), integration, and assignment. ^13^C NMR spectra were measured on a JEOL ECS-500 (125 MHz) spectrometer (see Appendix A). Chemical shifts were recorded in ppm from the solvent resonance employed as the internal standard (deuterochloroform at 77.0 ppm). High-resolution mass spectra were recorded by Thermo Fisher Scientific Exactive Orbitrap mass spectrometers. Infrared (IR) spectra were recorded on a JASCO FT/IR 4100 spectrometer. Single crystal X-ray diffraction data were collected at 173K on a Bruker SMART APEX II ultra CCD diffractometer with Cu Kα (λ = 1.54178) radiation and graphite monochromator. For thin-layer chromatography (TLC) analysis throughout this work, Merck precoated TLC plates (silica gel 60GF254 0.25 mm) were used. The products were purified by neutral column chromatography on silica gel (Kanto Chemical Co., Inc. silica gel 60N, Prod. No. 37560-84; Merck silica gel 60, Prod. No. 1.09385.9929). Visualization was accomplished by UV light (254 nm), anisaldehyde, KMnO_4_, and phosphomolybdic acid. In experiments that required dry solvents such as MeCN, toluene, and xylene (mixed isomers), these were distilled in prior to use.

### 3.2. General Procedure for 3-Amination of Indoles 1

A mixture of 1-(diacetoxyiodo)-2-butoxybenzene (94.6 mg, 0.24 mmol) and Ts_2_NH (104.1 mg, 0.32 mmol) in MeCN (2.0 mL) was stirred at room temperature for 30 min under argon atmosphere. Then, *N*-pivaloylindole (**1a**) (40.3 mg, 0.20 mmol) was added, and the solution was stirred at 40 °C for 7 h under argon atmosphere. The reaction mixture was concentrated under reduced pressure and the crude product was dissolved xylene (mixed isomers) (2.0 mL). Cu(MeCN)_4_BF_4_ (3.1 mg, 0.01 mmol) was added to the reaction mixture at room temperature, and further stirred at 130 °C for 1 h under argon atmosphere. Saturated NaHCO_3_ aqueous solution (10 mL) was added to the reaction mixture, and the product was extracted with AcOEt (15 mL × 3). The combined extracts were washed with brine (10 mL) and dried over Na_2_SO_4_. The organic phase was concentrated under reduced pressure and the crude product was purified by silica gel column chromatography (eluent: hexane/AcOEt = 5/1), to give the desired product **2a** (82.3 mg, 78% yield).

*4-Methyl-N-(1-pivaloyl-1H-indol-3-yl)-N-tosylbenzenesulfonamide* (**2a**). White solid, mp 214.0–214.5 °C, ^1^H NMR (500 MHz, CDCl_3_) δ 1.40 (s, 9H), 2.46 (s, 6H), 7.07 (d, *J* = 7.8 Hz, 1H), 7.14–7.19 (m, 1H), 7.30–7.37 (m, 1H), 7.32 (d, *J* = 8.2 Hz, 4H), 7.46 (s, 1H), 7.87 (d, *J* = 8.2 Hz, 4H), 8.46 (d, *J* = 8.3 Hz, 1H). ^13^C NMR (125 MHz, CDCl_3_) δ 21.7 (2C), 28.5 (3C), 41.3, 115.8, 117.3, 118.6, 124.1, 126.0, 126.8, 127.7, 128.6 (4C), 129.6 (4C), 135.7, 136.2 (2C), 145.2 (2C), 176.7. IR (neat) 1702, 1374, 1357, 1316, 1165 cm^–1^. MS (ESI) calcd for C_27_H_29_N_2_O_5_S_2_ [M + H]^+^ 525.1512, found 525.1501. Crystal data for 2a: Formula C_27_H_28_N_2_O_5_S_2_, colorless, crystal dimensions 0.30 × 0.20 × 0.20 mm^3^, Triclinic, space group P −1, *a* = 9.475(2) Å, *b* = 10.144(2) Å, *c* = 13.660(3) Å, α = 98.118(3) °, β = 91.348(3) °, γ = 100.871(3) °, *V* = 1274.7(5) Å^3^, *Z* = 2, ρ*_calc_* = 1.367 g cm^−3^, F(000) = 552, μ(MoKα) = 0.250 mm^−1^, *T* = 173 K. 7413 reflections collected, 5602 independent reflections with *I* > 2σ(*I*) (2θ_max_ = 27.572°), and 330 parameters were used for the solution of the structure. The non-hydrogen atoms were refined anisotropically. *R*_1_ = 0.0508 and *wR*_2_ = 0.1031. GOF = 1.015. Crystallographic data (excluding structure factors) for the structure reported in this paper have been deposited with the Cambridge Crystallographic Data Centre as supplementary publication no. CCDC-1860598. Copies of the data can be obtained free of charge on application to CCDC, 12 Union Road, Cambridge CB2 1EZ, UK [Fax: int. code + 44(1223)336-033; E-mail: deposit@ccdc.cam.ac.uk].

*N**-(5-Chloro-1-pivaloyl-1H-indol-3-yl)-4-methyl-N-tosylbenzenesulfonamide* (**2b**): White solid, mp 205.5–206.0 °C, ^1^H NMR (500 MHz, CDCl_3_) δ 1.41 (s, 9H), 2.48 (s, 6H), 6.81 (d, *J* = 2.0 Hz, 1H), 7.27 (d, *J* = 8.9, 2.0 Hz, 1H), 7.33 (d, *J* = 8.5 Hz, 4H), 7.52 (s, 1H), 7.85 (d, *J* = 8.5 Hz, 4H), 8.38 (d, *J* = 8.9 Hz, 1H). ^13^C NMR (125 MHz, CDCl_3_) δ 21.7 (2C), 28.4 (3C), 41.3, 115.2, 118.2, 118.4, 126.2, 128.0, 128.6 (4C), 128.7, 129.7 (4C), 130.0, 133.9, 136.0 (2C), 145.6 (2C), 176.6. IR (neat) 1708, 1379, 1351, 1307, 1163, 1085 cm^–1^. MS (ESI) calcd for C_27_H_28_ClN_2_O_5_S_2_ [M + H]^+^ 559.1123, found 559.1119.

*N**-(Phenylsulfonyl)-N-(1-pivaloyl-1H-indol-3-yl)benzenesulfonamide* (**2c**). White solid, mp 182.0–183.0 °C, ^1^H NMR (500 MHz, CDCl_3_) δ 1.39 (s, 9H), 7.01 (d, *J* = 8.0 Hz, 1H), 7.12–7.18 (m, 1H), 7.31–7.37 (m, 1H), 7.48 (s, 1H), 7.50–7.56 (m, 4H), 7.67 (t, *J* = 7.5 Hz, 2H), 7.97–8.02 (m, 4H), 8.47 (d, *J* = 8.5 Hz, 1H). ^13^C NMR (125 MHz, CDCl_3_) δ 28.5 (3C), 41.3, 115.5, 117.3, 118.5, 124.2, 126.1, 126.6, 127.7, 128.5 (4C), 129.0 (4C), 134.1 (2C), 135.6, 139.0 (2C), 176.7. IR (neat) 1704, 1381, 1344, 1315, 1159 cm^–1^. MS (ESI) calcd for C_25_H_25_N_2_O_5_S_2_ [M + H]^+^ 497.1199, found 497.1205.

*4-Methyl-N-(methylsulfonyl)-N-(1-pivaloyl-1H-indol-3-yl)benzenesulfonamide* (**2d**). White solid, mp 221.5–22.0 °C, ^1^H NMR (500 MHz, CDCl_3_) δ 1.44 (s, 9H), 2.43 (s, 3H), 3.56 (s, 3H), 7.23–7.30 (m, 3H), 7.30–7.34 (m, 1H), 7.34–7.40 (m, 1H), 7.62 (s, 1H), 7.79 (d, *J* = 8.3 Hz, 2H), 8.48 (d, *J* = 8.3 Hz, 1H). ^13^C NMR (125 MHz, CDCl_3_) δ 21.7, 28.5 (3C), 41.4, 44.0, 115.2, 117.4, 118.2, 124.4, 126.2, 126.7, 127.2, 128.8 (2C), 129.6 (2C), 135.1, 135.7, 145.6, 176.7. IR (neat) 1704, 1366, 1349, 1317, 1157 cm^–1^. MS (ESI) calcd for C_21_H_2__5_N_2_O_5_S_2_ [M + H]^+^ 449.1199, found 449.1198.

*N**-(1-Pivaloyl-1H-indol-3-yl)-N-(propylsulfonyl)propane-1-sulfonamide* (**2e**). White solid, mp 190.0–191.0 °C, ^1^H NMR (500 MHz, CDCl_3_, 60 °C) δ 1.07 (t, *J* = 7.3 Hz, 6H), 1.52 (s, 9H), 1.88–2.01 (m, 4H), 3.37–3.55 (m, 2H), 3.55–3.75 (m, 2H), 7.31–7.41 (m, 2H), 7.62 (d, *J* = 7.5 Hz, 1H), 7.88 (s, 1H), 8.50 (d, *J* = 8.0 Hz, 1H). ^13^C NMR (125 MHz, CDCl_3_, 60 °C) δ 12.8 (2C), 17.1 (2C), 28.7 (3C), 41.5, 57.7 (2C), 115.5, 117.6, 118.5, 124.5, 126.2, 126.9, 127.2, 135.9, 176.7. IR (neat) 1703, 1370, 1347, 1316, 1151 cm^–1^. MS (ESI) calcd for C_19_H_29_N_2_O_5_S_2_ [M + H]^+^ 429.1512, found 429.1512.

*N-(Methylsulfonyl)-N-(1-pivaloyl-1H-indol-3-yl)methanesulfonamide* (**2f**). White solid, mp 205.0–207.0 °C, ^1^H NMR (500 MHz, CDCl_3_) δ 1.54 (s, 9H), 3.44 (s, 6H), 7.35–7.40 (m, 1H), 7.40–7.45 (m, 1H), 7.58–7.63 (m, 1H), 7.94 (s, 1H), 8.53 (d, *J* = 8.3 Hz, 1H). ^13^C NMR (125 MHz, CDCl_3_) δ 28.6 (3C), 41.5, 42.9 (2C), 114.6, 117.7, 118.0, 124.7, 126.3, 126.4, 127.1, 135.8, 176.8. IR (neat) 1703, 1349, 1317, 1159 cm^–1^. MS (ESI) calcd for C_15_H_20_ClN_2_O_5_S_2_ [M + Cl]^−^ 407.0508, found 407.0515.

*4-Methyl-N-(5-methyl-1-pivaloyl-1H-indol-3-yl)-N-tosylbenzenesulfonamide* (**2g**). White solid, mp 217.0–217.5 °C, ^1^H NMR (500 MHz, CDCl_3_) δ 1.39 (s, 9H), 2.28 (s, 3H), 2.47 (s, 6H), 6.71–6.73 (m, 1H), 7.12–7.16 (m, 1H), 7.32 (d, *J* = 8.2 Hz, 4H), 7.42 (s, 1H), 8.87 (d, *J* = 8.2 Hz, 4H), 8.31 (d, *J* = 8.6 Hz, 1H). ^13^C NMR (125 MHz, CDCl_3_) δ 21.2, 21.7 (2C), 28.5 (3C), 41.2, 115.6, 116.9, 118.4, 126.9, 127.4, 127.6, 128.7 (4C), 129.5 (4C), 133.8, 133.9, 136.3 (2C), 145.2 (2C), 176.6. IR (neat) 1701, 1379, 1353, 1308, 1156 cm^–1^. MS (ESI) calcd for C_28_H_31_N_2_O_5_S_2_ [M + H]^+^ 539.1669, found 539.1672.

*N**-(5-Fluoro-1-pivaloyl-1H-indol-3-yl)-4-methyl-N-tosylbenzenesulfonamide* (**2h**). White solid, mp 214.5–215.2 °C, ^1^H NMR (500 MHz, CDCl_3_) δ 1.40 (s, 9H), 2.47 (s, 6H), 6.66 (dd, *J* = 8.5, 2.5 Hz, 1H), 7.05 (td, *J* = 9.0, 2.5 Hz, 1H), 7.33 (d, *J* = 8.5 Hz, 4H), 7.51 (s, 1H), 7.86 (d, *J* = 8.5 Hz, 4H), 8.43 (dd, *J* = 9.0, 4.5 Hz, 1H). ^13^C NMR (125 MHz, CDCl_3_) δ 21.7 (2C), 28.5 (3C), 41.3, 104.3 (d, *J*_C–F_ = 25.0 Hz), 113.9 (d, *J*_C–F_ = 25.0 Hz), 115.5 (d, *J*_C–F_ = 3.5 Hz), 118.6 (d, *J*_C–F_ = 9.5 Hz), 128.1 (d, *J*_C–F_ = 9.5 Hz), 128.5 (4C), 129.0, 129.7 (4C), 131.9, 136.1 (2C), 145.5 (2C), 159.9 (d, *J*_C–F_ = 240.8 Hz), 176.5. ^19^F NMR (471 MHz, CDCl_3_) δ −117.9. IR (neat) 1708, 1376, 1359, 1311, 1237, 1169 cm^–1^. MS (ESI) calcd for C_27_H_28_FN_2_O_5_S_2_ [M + H]^+^ 543.1418, found 543.1422.

*N**-(5-Bromo-1-pivaloyl-1H-indol-3-yl)-4-methyl-N-tosylbenzenesulfonamide* (**2i**). White solid, mp 215.0–216.0 °C, ^1^H NMR (500 MHz, CDCl_3_) δ 1.41 (s, 9H), 2.48 (s, 6H), 6.09 (d, *J* = 1.8 Hz, 1H), 7.31–7.35 (m, 4H), 7.40 (dd, *J* = 9.0, 1.8 Hz, 1H), 7.51 (s, 1H), 7.82–7.86 (m, 4H), 8.32 (d, *J* = 9.0 Hz, 1H). ^13^C NMR (125 MHz, CDCl_3_) δ 21.7 (2C), 28.4 (3C), 41.3, 115.0, 117.7, 118.7, 121.3, 128.4, 128.50, 128.55 (4C), 128.8, 129.7 (4C), 134.3, 136.0 (2C), 145.6 (2C), 176.6. IR (neat) 1703, 1382, 1348, 1306, 1164, 659 cm^–1^. MS (ESI) calcd for C_27_H_28_BrN_2_O_5_S_2_ [M + H]^+^ 603.0618, found 603.0620.

*N**-(5-Methoxy-1-pivaloyl-1H-indol-3-yl)-4-methyl-N-tosylbenzenesulfonamide* (**2j**). White solid, mp 220.0–221.0 °C, ^1^H NMR (500 MHz, CDCl_3_) δ 1.39 (s, 9H), 2.46 (s, 6H), 3.62 (s, 3H), 6.37 (d, *J* = 2.6 Hz, 1H), 6.92 (dd, *J* = 9.1, 2.6 Hz, 1H), 7.32 (d, *J* = 8.2 Hz, 4H), 7.43 (s, 1H), 7.88 (d, *J* = 8.2 Hz, 4H), 8.34 (d, *J* = 9.1 Hz, 1H). ^13^C NMR (125 MHz, CDCl_3_) δ 21.7 (2C), 28.6 (3C), 41.2, 55.2, 100.1, 115.4, 115.6, 118.3, 127.8, 128.0, 128.6 (4C), 129.6 (4C), 130.2, 136.4 (2C), 145.2 (2C), 156.7, 176.4. IR (neat) 1701, 1380, 1358, 1312, 1162, 1086 cm^–1^. MS (ESI) calcd for C_28_H_31_N_2_O_6_S_2_ [M + H]^+^ 555.1618, found 555.1621.

*3-((4-Methyl-N-tosylphenyl)sulfonamido)-1-pivaloyl-1H-indol-5-yl pivalate* (**2k**): White solid, mp 225.0 °C (dec.), ^1^H NMR (500 MHz, CDCl_3_) δ 1.35 (s, 9H), 1.40 (s, 9H), 2.45 (s, 6H), 6.64 (d, *J* = 2.5 Hz, 1H), 7.02 (dd, *J* = 9.0, 2.5 Hz, 1H), 7.33 (d, *J* = 8.3 Hz, 4H), 7.48 (s, 1H), 7.88 (d, *J* = 8.3 Hz, 4H), 8.46 (d, *J* = 9.0 Hz, 1H). ^13^C NMR (125 MHz, CDCl_3_) δ 21.7 (2C), 27.1 (3C), 28.5 (3C), 38.9, 41.3, 113.3, 115.7, 118.1, 120.0, 127.6, 128.58 (4C), 128.63, 129.7 (4C), 133.2, 136.2 (2C), 145.3 (2C), 147.7, 176.5, 176.8. IR (neat) 1750, 1708, 1377, 1358, 1311, 1167, 1119 cm^–1^. MS (ESI) calcd for C_32_H_37_N_2_O_7_S_2_ [M + H]^+^ 625.2037, found 625.2937.

*Methyl 3-((4-methyl-N-tosylphenyl)sulfonamido)-1-pivaloyl-1H-indole-5-carboxylate* (**2l**). White solid, mp 221.0–221.5 °C, ^1^H NMR (500 MHz, CDCl_3_) δ 1.41 (s, 9H), 2.46 (s, 6H), 3.89 (s, 3H), 7.33 (d, *J* = 8.3 Hz, 4H), 7.54 (s, 1H), 7.63 (d, *J* = 1.5 Hz, 1H), 7.86 (d, *J* = 8.3 Hz, 4H), 8.02 (d, *J* = 8.9, 1.5 Hz, 1H), 8.50 (d, *J* = 8.9 Hz, 1H). ^13^C NMR (125 MHz, CDCl_3_) δ 21.7 (2C), 28.4 (3C), 41.4, 52.0, 116.2, 117.1, 120.7, 126.1, 126.6, 127.2, 128.6 (4C), 128.8, 129.7 (4C), 136.0 (2C), 138.1, 145.5 (2C), 166.7, 176.7. IR (neat) 1717, 1379, 1360, 1313, 1165 cm^–1^. MS (ESI) calcd for C_29_H_31_N_2_O_7_S_2_ [M + H]^+^ 583.1567, found 583.1571.

*N**-(5-Cyano-1-pivaloyl-1H-indol-3-yl)-4-methyl-N-tosylbenzenesulfonamide* (**2m**). White solid, mp 214.0–215.0 °C, ^1^H NMR (500 MHz, CDCl_3_) δ 1.42 (s, 9H), 2.49 (s, 6H), 7.17 (d, *J* = 1.5 Hz, 1H), 7.34 (d, *J* = 8.0 Hz, 4H), 7.55 (dd, *J* = 9.0, 1.5 Hz, 1H), 7.64 (s, 1H), 7.83 (d, *J* = 8.0 Hz, 4H), 8.55 (d, *J* = 8.5 Hz, 1H). ^13^C NMR (125 MHz, CDCl_3_) δ 21.7 (2C), 28.2 (3C), 41.5, 107.6, 115.5, 118.2, 118.8, 123.4, 126.9, 128.5 (4C), 128.7, 129.5, 129.8 (4C), 135.7 (2C), 137.2, 145.8 (2C), 176.6. IR (neat) 2221, 1715, 1363, 1348, 1308, 1169, 1086 cm^–1^. MS (ESI) calcd for C_28_H_28_N_3_O_5_S_2_ [M + H]^+^ 550.1465, found 550.1470.

*N**-(5-(1,3-Dioxoisoindolin-2-yl)-1-pivaloyl-1H-indol-3-yl)-4-methyl-N-tosylbenzenesulfonamide* (**2n**). White solid, mp 227.5–228.0 °C, ^1^H NMR (500 MHz, CDCl_3_) δ 1.39 (s, 9H), 2.44 (s, 6H), 7.23 (d, *J* = 2.0 Hz, 1H), 7.38 (d, *J* = 8.3 Hz, 4H), 7.42 (dd, *J* = 9.0, 2.0 Hz, 1H), 7.47 (s, 1H), 7.80 (dd, *J* = 5.5, 3.0 Hz, 2H), 7.92 (d, *J* = 8.3 Hz, 4H), 7.96 (dd, *J* = 5.5, 3.0 Hz, 2H), 8.61 (d, *J* = 9.0 Hz, 1H). ^13^C NMR (125 MHz, CDCl_3_) δ 21.7 (2C), 28.4 (3C), 41.3, 115.6, 117.5, 118.0, 123.6 (2C), 124.7, 127.3, 127.9, 128.6 (4C), 128.9, 129.8 (4C), 131.8 (2C), 134.3 (2C), 134.9, 136.0 (2C), 145.3 (2C), 167.1 (2C), 176.6. IR (neat) 1727, 1705, 1375, 1355, 1308, 1167, 1080 cm^–1^. MS (ESI) calcd for C_35_H_32_N_3_O_7_S_2_ [M + H]^+^ 670.1676, found 670.1678.

*4-Methyl-N-(6-methyl-1-pivaloyl-1H-indol-3-yl)-N-tosylbenzenesulfonamide* (**2o**). White solid, mp 203.5–204.2 °C, ^1^H NMR (500 MHz, CDCl_3_) δ 1.38 (s, 9H), 2.45 (s, 3H), 2.46 (s, 6H), 6.95 (d, *J* = 8.0 Hz, 1H), 7.01 (d, *J* = 8.0 Hz, 1H), 7.29–7.35 (m, 4H), 7.37 (s, 1H), 7.84–7.89 (m, 4H), 8.32 (s, 1H). ^13^C NMR (125 MHz, CDCl_3_) δ 21.7 (2C), 21.9, 28.4 (3C), 41.3, 115.8, 117.4, 118.2, 124.6, 125.6, 127.0, 128.6 (4C), 129.6 (4C), 136.1, 136.26, 136.28 (2C), 145.2 (2C), 176.9. IR (neat) 1704, 1371, 1339, 1313, 1163 cm^–1^. MS (ESI) calcd for C_28_H_30_ClN_2_O_5_S_2_ [M + Cl]^–^ 573.1290, found 573.1303.

*N-(6-Fluoro-1-pivaloyl-1H-indol-3-yl)-4-methyl-N-tosylbenzenesulfonamide* (**2p**). White solid, mp 215.0–215.5 °C, ^1^H NMR (500 MHz, CDCl_3_) δ 1.40 (s, 9H), 2.46 (s, 6H), 6.89–7.02 (m, 2H), 7.32 (d, *J* = 8.0 Hz, 4H), 7.46 (s, 1H), 7.85 (d, *J* = 8.0 Hz, 4H), 8.22 (dd, *J* = 10.3, 2.0 Hz, 1H). ^13^C NMR (125 MHz, CDCl_3_) δ 21.7 (2C), 28.4 (3C), 41.3, 104.7 (d, *J*_C–F_ = 29.8 Hz), 112.6 (d, *J*_C–F_ = 23.9 Hz), 115.7, 119.4 (d, *J*_C–F_ = 9.5 Hz), 123.1, 127.8 (d, *J*_C–F_ = 3.5 Hz), 128.5 (4C), 129.7 (4C), 135.7 (d, *J*_C–F_ = 13.1 Hz), 136.1 (2C), 145.4 (2C), 161.7 (d, *J*_C–F_ = 240.9 Hz), 176.6. ^19^F NMR (471 MHz, CDCl_3_) δ –115.1. IR (neat) 1707, 1382, 1345, 1315, 1230, 1162 cm^–1^. MS (ESI) calcd for C_27_H_27_ClFN_2_O_5_S_2_ [M + Cl]^–^ 577.1039, found 577.1055.

*N**-(6-Chloro-1-pivaloyl-1H-indol-3-yl)-4-methyl-N-tosylbenzenesulfonamide* (**2q**). White solid, mp 212.5–213.0 °C, ^1^H NMR (500 MHz, CDCl_3_) δ 1.40 (s, 9H), 2.47 (s, 6H), 6.94 (d, *J* = 8.6 Hz, 1H), 7.14 (dd, *J* = 8.6, 1.7 Hz, 1H), 7.30–7.35 (m, 4H), 7.46 (s, 1H), 7.82–7.87 (m, 4H), 8.54 (d, *J* = 1.7 Hz, 1H). ^13^C NMR (125 MHz, CDCl_3_) δ 21.7 (2C), 28.4 (3C), 41.3, 115.7, 117.6, 119.4, 124.8, 125.4, 128.1, 128.5 (4C), 129.7 (4C), 132.1, 135.8, 136.1 (2C), 145.4 (2C), 176.6. IR (neat) 1706, 1381, 1360, 1337, 1160, 1083 cm^–1^. MS (ESI) calcd for C_27_H_28_ClN_2_O_5_S_2_ [M + H]^+^ 559.1123, found 559.1123.

*N-(6-Bromo-1-pivaloyl-1H-indol-3-yl)-4-methyl-N-tosylbenzenesulfonamide* (**2r**). White solid, mp 214.0–214.8 °C, ^1^H NMR (500 MHz, CDCl_3_) δ 1.39 (s, 9H), 2.47 (s, 6H), 6.90 (d, *J* = 8.5 Hz, 1H), 7.29 (dd, *J* = 8.5, 1.5 Hz, 1H), 7.32 (d, *J* = 8.5 Hz, 4H), 7.44 (s, 1H), 7.85 (d, *J* = 8.5 Hz, 4H), 8.71 (d, *J* = 1.5 Hz, 1H). ^13^C NMR (125 MHz, CDCl_3_) δ 21.7 (2C), 28.3 (3C), 41.3, 115.7, 119.7, 119.9, 120.4, 125.7, 127.5, 128.0, 128.5 (4C), 129.7 (4C), 136.06 (2C), 136.11, 145.4 (2C), 176.6. IR (neat) 1705, 1380, 1336, 1308, 1159, 658 cm^–1^. MS (ESI) calcd for C_27_H_27_BrClN_2_O_5_S_2_ [M + Cl]^–^ 637.0239, found 637.0256.

*N-(6-Methoxy-1-pivaloyl-1H-indol-3-yl)-4-methyl-N-tosylbenzenesulfonamide* (**2s**). White solid, mp 201.0–201.8 °C, ^1^H NMR (500 MHz, CDCl_3_) δ 1.40 (s, 9H), 2.46 (s, 6H), 3.86 (s, 3H), 6.80 (dd, *J* = 8.5, 2.3 Hz, 1H), 6.90 (d, *J* = 8.5 Hz, 1H), 7.29–7.34 (m, 4H), 7.36 (s, 1H), 7.84–7.89 (m, 4H), 8.07 (d, *J* = 2.3 Hz, 1H). ^13^C NMR (125 MHz, CDCl_3_) δ 21.7 (2C), 28.4 (3C), 41.3, 55.6, 100.8, 114.1, 115.9, 119.1, 120.5, 126.2, 128.5 (4C), 129.6 (4C), 136.3 (2C), 136.7, 145.2 (2C), 159.0, 177.1. IR (neat) 1703, 1378, 1346, 1312, 1286, 1162 cm^–1^. MS (ESI) calcd for C_28_H_30_ClN_2_O_5_S_2_ [M + Cl]^–^ 589.1239, found 589.1257.

*N-(4-Fluoro-1-pivaloyl-1H-indol-3-yl)-4-methyl-N-tosylbenzenesulfonamide* (**2t**). White solid, mp 220.0–222.6 °C, ^1^H NMR (500 MHz, CDCl_3_) δ 1.42 (s, 9H), 2.46 (s, 6H), 6.85 (dd, *J* = 10.0, 8.0 Hz, 1H), 7.24–7.30 (m, 1H), 7.30–7.34 (m, 4H), 7.50 (s, 1H), 7.84–7.88 (m, 4H), 8.28 (d, *J* = 8.5 Hz, 1H). ^13^C NMR (125 MHz, CDCl_3_) δ 21.7 (2C), 28.4 (3C), 41.4, 110.2 (d, *J*_C–F_ = 19.0 Hz), 112.9, 113.3 (d, *J*_C–F_ = 3.6 Hz), 115.7 (d, *J*_C–F_ = 17.9 Hz), 126.7 (d, *J*_C–F_ = 7.3 Hz), 128.0, 128.6 (4C), 129.5 (4C), 136.1(2C), 137.7 (d, *J*_C–F_ = 7.1 Hz), 145.1 (2C), 154.7 (d, *J*_C–F_ = 249.1 Hz), 176.7. ^19^F NMR (471 MHz, CDCl_3_) δ –122.8. IR (neat) 1707, 1375, 1359, 1309, 1251, 1176 cm^–1^. MS (ESI) calcd for C_27_H_27_ClFN_2_O_5_S_2_ [M + Cl]^–^ 577.1039, found 577.1053.

*4-Methyl-N-(7-methyl-1-pivaloyl-1H-indol-3-yl)-N-tosylbenzenesulfonamide* (**2u**). White solid, mp 190.0–191.0 °C, ^1^H NMR (500 MHz, CDCl_3_) δ 1.41 (s, 9H), 2.30 (s, 3H), 2.45 (s, 6H), 6.94 (dd, *J* = 7.5, 1.5 Hz, 1H), 7.06–7.12 (m, 2H), 7.20 (s, 1H), 7.29–7.33 (m, 4H), 7.84–7.89 (m, 4H). ^13^C NMR (125 MHz, CDCl_3_) δ 21.2, 21.7 (2C), 28.9 (3C), 42.1, 114.8, 116.6, 123.8, 125.4, 127.5, 128.00, 128.03, 128.6 (4C), 129.5 (4C), 134.9, 136.3 (2C), 145.1 (2C), 178.2. IR (neat) 1715, 1377, 1361, 1303, 1171 cm^–1^. MS (APCI) calcd for C_28_H_31_N_2_O_5_S_2_ [M + H]^+^ 539.1669, found 539.1670.

*N**-(5,6-Dichloro-1-pivaloyl-1H-indol-3-yl)-4-methyl-N-tosylbenzenesulfonamide* (**2v**). White solid, mp 223.5–224.2 °C, ^1^H NMR (500 MHz, CDCl_3_) δ 1.41 (s, 9H), 2.48 (s, 6H), 6.87 (s, 1H), 7.32–7.36 (m, 4H), 7.52 (s, 1H), 7.82–7.86 (m, 4H), 8.65 (s, 1H). ^13^C NMR (125 MHz, CDCl_3_) δ 21.7 (2C), 28.3 (3C), 41.3, 115.0, 119.1, 119.5, 126.4, 128.5 (5C), 129.0, 129.8 (4C), 130.1, 134.0, 135.9 (2C), 145.7 (2C), 176.4. IR (neat) 1710, 1382, 1361, 1330, 1158, 1084 cm^–1^. MS (ESI) calcd for C_27_H_27_Cl_2_N_2_O_5_S_2_ [M + H]^+^ 593.0733, found 593.0732.

*4-Methyl-N-(2-methyl-1-pivaloyl-1H-indol-3-yl)-N-tosylbenzenesulfonamide* (**2w**). White solid, mp 158.0–159.0 °C, ^1^H NMR (500 MHz, CDCl_3_) δ 1.34 (s, 9H), 1.79 (s, 3H), 2.45 (s, 6H), 6.99–7.06 (m, 2H), 7.12–7.18 (m, 1H), 7.21 (d, *J* = 8.0 Hz, 1H), 7.30 (d, *J* = 8.5 Hz, 4H), 7.85 (d, *J* = 8.5 Hz, 4H). ^13^C NMR (125 MHz, CDCl_3_) δ 10.8, 21.6 (2C), 28.0 (3C), 44.4, 110.5, 111.7, 118.7, 121.7, 122.8, 125.8, 128.6 (4C), 129.5 (4C), 133.8, 136.7 (2C), 137.9, 145.0 (2C), 185.8. IR (neat) 1710, 1384, 1360, 1313, 1165, 1084 cm^–1^. MS (APCI) calcd for C_28_H_31_N_2_O_5_S_2_ [M + H]^+^ 539.1669, found 539.1671.

*N**-(1-Benzoyl-1H-indol-3-yl)-4-methyl-N-tosylbenzenesulfonamide* (**2x**). White solid, mp 214.0–215.0 °C, ^1^H NMR (500 MHz, CDCl_3_) δ 2.47 (s, 6H), 7.10 (d, *J* = 8.0 Hz, 1H), 7.12 (s, 1H), 7.19–7.24 (m, 1H), 7.32 (d, *J* = 8.5 Hz, 4H), 7.36–7.41 (m, 1H), 7.45–7.51 (m, 2H), 7.59–7.65 (m, 3H), 7.87 (d, *J* = 8.5 Hz, 4H), 8.37 (d, *J* = 8.0 Hz, 1H). ^13^C NMR (125 MHz, CDCl_3_) δ 21.7 (2C), 116.2, 116.3, 119.0, 1124.5, 125.8, 128.0, 128.59 (4C), 128.64 (2C), 129.3, 129.4 (2C), 129.6 (4C), 132.5, 133.4, 135.0, 136.2 (2C), 145.2 (2C), 168.1. IR (neat) 1694, 1376, 1359, 1325, 1166 cm^–1^. MS (ESI) calcd for C_29_H_25_N_2_O_5_S_2_ [M + H]^+^ 545.1199, found 545.1205.

*4-Methyl-N-tosyl-N-(1-tosyl-1H-indol-3-yl)benzenesulfonamide* (**2y**). White solid, mp 182.0–183.0 °C, ^1^H NMR (500 MHz, CDCl_3_) δ 2.35 (s, 3H), 2.44 (s, 6H), 7.04 (d, *J* = 8.0 Hz, 1H), 7.08–7.13 (m, 1H), 7.23–7.30 (m, 7H), 7.43 (s, 1H), 7.70–7.77 (m, 6H), 7.92 (d, *J* = 8.5 Hz, 1H). ^13^C NMR (125 MHz, CDCl_3_) δ 21.5, 21.7 (2C), 113.5, 116.8, 119.4, 123.9, 125.4, 126.9 (2C), 128.2, 128.3, 128.5 (4C), 129.5 (4C), 130.0 (2C), 133.7, 134.5, 136.0 (2C), 145.3 (2C), 145.5. IR (neat) 1595, 1372, 1355, 1283, 1168, 1086 cm^–1^. MS (ESI) calcd for C_29_H_27_N_2_O_6_S_3_ [M + H]^+^ 595.1026, found 595.1027.

## 4. Conclusions

In conclusion, we developed a 3-amination of indole derivatives using a hypervalent iodine(III) compound that promotes the formation of indolyl(aryl)iodonium imides. This reaction was followed by a copper-catalyzed oxidative C–N coupling reaction to obtain the 3-amino indole derivatives regioselectively. The *o*-alkoxy groups on (diacetoxy)iodoarenes resulted in higher reactivity than the *o*-alkyl groups in the formation of indolyl(aryl)iodonium imides (first step). In addition, the 3-amino indole derivatives were given as a single isomer product by a decomposition of 2-amino indole derivatives under copper-catalyzed reaction conditions (second step). The design of substituted iodoarene as a high-performance hypervalent iodine compounds and the combined use of hypervalent iodine compound and a transition-metal catalyst for the unique transformation are underway in our laboratory.

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
