# Peer review of "Cu-Catalyzed Oxidative 3-Amination of Indoles via Formation of Indolyl(aryl)iodonium Imides Using o-Substituted (Diacetoxyiodo)arene as a High-Performance Hypervalent Iodine Compound"

_molecules, 2019, doi:10.3390/molecules24061147_

Reviewer 1 Report

The manuscript authored by Watanabe and Moriyama reports a novel amination protocol of indoles by two consecutive steps: the formation of indolyl(aryl)iodonium imides using o-substituted (diacetoxyiodo)arene and a following Cu-catalyzed oxidation. Even though the same research group reports several articles in the field, this specific application is original and deserves publication, also considering the great amount of work performed. However, it is not clear why it is not possible to perform one-pot procedures and consequently the necessity to change the solvent should be clearly explained. I have also some concerns about the style. The manuscript is written is a rather obscure manner. I suggest to make some efforts to improve the comprehension of the work, modifying the text and improving the quality of the schemes for examples adding colors.

Author Response

The manuscript authored by Watanabe and Moriyama reports a novel amination protocol of indoles by two consecutive steps: the formation of indolyl(aryl)iodonium imides using o-substituted (diacetoxyiodo)arene and a following Cu-catalyzed oxidation. Even though the same research group reports several articles in the field, this specific application is original and deserves publication, also considering the great amount of work performed.

Response: Thank you very much for your recommendation for publication in Molecules. We revised the manuscript for publication in Molecules based on talking the reviewer 1’s helpful comments.

However, it is not clear why it is not possible to perform one-pot procedures and consequently the necessity to change the solvent should be clearly explained.

We added “It is important to use MeCN as a solvent in the formation of indolyl(aryl)iodonium imides (in the first step) and toluene-related solvent (i.e. toluene and xylene) in the copper-catalyzed C–N coupling reaction (in the second step) to proceed each reaction efficiently.” at line 73 to explain the necessity to change the solvent.

I have also some concerns about the style. The manuscript is written is a rather obscure manner. I suggest to make some efforts to improve the comprehension of the work, modifying the text and improving the quality of the schemes for examples adding colors.

We improved our manuscript by adding colors in the schemes. 

Reviewer 2 Report

Manuscript by Moriyama describes the Cu-catalyzed 3-amination of indoles via an hypervalent  iodine(III) intermediate. 

This ms. is a spin-off of a JOC published in 2018 (Ref21) with substantially no original result.  The extension to other potential dummy ligands (this point was already illustrated in ref 21, see Table 1) does not bring any improvement (scope or yield). 

In this referee's opinion, manuscript lacks elements of novelty to warrant publication.

Author Response

Manuscript by Moriyama describes the Cu-catalyzed 3-amination of indoles via an hypervalent  iodine(III) intermediate. 

This ms. is a spin-off of a JOC published in 2018 (Ref21) with substantially no original result.  The extension to other potential dummy ligands (this point was already illustrated in ref 21, see Table 1) does not bring any improvement (scope or yield). 

In this referee's opinion, manuscript lacks elements of novelty to warrant publication.

Response: Thank you very much for your great effort for reviewing our manuscript. The manuscript describes the superiority of o-alkoxy(diacetoxyiodo)arenes as a high-performance hypervalent iodine compound compered to o-alkyl(diacetoxyiodo) arenes for 3-amination of indole derivatives. Furthermore, we designed 2,6-dimethoxy(diacetoxyiodo)benzene as a o-dialkoxy(diacetoxyiodo)arene for the present reaction, and the 2,6-dimethoxy(diacetoxyiodo)benzene improved the yield of 3-aminoindole derivatives (e.g. 2e, 2h, 2i, 2k, 2l, 2m, 2n, 2t, and 2u). Therefore, we believe that this results is different the previous results, and has some originalities and improvements for 3-amination of indole derivatives.   

Reviewer 3 Report

This manuscript reports a usual addition / update to the toolbox for C-3 amination of indoles, a generally important class of reaction. Comparison with previous methods were well made while describing the features of each system. The series of hypervalent iodine reagents tested were informative and the rationale for each choice was clear by the experiment / trends observed. Also good range of substrates examined. Excellent characterization data.

Only minor issues:

- formatting is off in the paragraph beginning at line 82.

- which xylenes were used? ortho, meta, or para?

Author Response

This manuscript reports a usual addition / update to the toolbox for C-3 amination of indoles, a generally important class of reaction. Comparison with previous methods were well made while describing the features of each system. The series of hypervalent iodine reagents tested were informative and the rationale for each choice was clear by the experiment / trends observed. Also good range of substrates examined. Excellent characterization data.

Only minor issues:

Response: Thank you very much for your recommendation for publication in Molecules. We revised the manuscript for publication in Molecules based on talking the reviewer 3’s helpful comments. 

- formatting is off in the paragraph beginning at line 82.

We checked the formatting of the paragraph at line 82. However, the formatting in our file is correct.  

- which xylenes were used? ortho, meta, or para?

We used an isomer mixed xylene solvent in the present reaction. We added “xylene (mixed isomer)” in General procedure to clear the protocol of experiment.

Round  2

Reviewer 1 Report

The manuscript has been extensively revised. In my opinion it can be accepted for publication

Reviewer 2 Report

Publication is recommended